# Manipulation of Auxin Signaling by Smut Fungi during Plant Colonization

**DOI:** 10.3390/jof9121184

**Published:** 2023-12-11

**Authors:** Nithya Nagarajan, Mamoona Khan, Armin Djamei

**Affiliations:** Department of Plant Pathology, Institute of Crop Science and Resource Conservation (INRES), University of Bonn, 53115 Bonn, Germany; nnagaraj@uni-bonn.de (N.N.); khanma@uni-bonn.de (M.K.)

**Keywords:** smut fungi, auxin signaling, biotrophic interactions, effectors, TOPLESS

## Abstract

A common feature of many plant-colonizing organisms is the exploitation of plant signaling and developmental pathways to successfully establish and proliferate in their hosts. Auxins are central plant growth hormones, and their signaling is heavily interlinked with plant development and immunity responses. Smuts, as one of the largest groups in basidiomycetes, are biotrophic specialists that successfully manipulate their host plants and cause fascinating phenotypes in so far largely enigmatic ways. This review gives an overview of the growing understanding of how and why smut fungi target the central and conserved auxin growth signaling pathways in plants.

## 1. Smut Fungi, Their Development, and Disease Symptoms

Smut fungi form beside rusts, the second-largest group of basidiomycetes, and comprise more than 1500 species [1]. While single smut species are specialists, in sum, their host range is very broad. They infect as biotrophic fungi, mainly monocots, including important crops such as maize, wheat, barley, and sugarcane, resulting in economically relevant diseases if not prevented in time. The life cycle of smut fungi is notable because it includes both sexual and asexual reproduction. They can be grown in axenic culture and have a biotrophic lifestyle during host colonization. Due to their ease of propagation, genetic accessibility, agronomic relevance, and intriguing developmental and distinguishing disease symptoms, they are widely studied by biologists [2]. Studying smut fungi can provide fascinating insights into plant–pathogen interactions. It broadens our understanding of how plants can be manipulated and supports the development of disease prevention strategies. Most of our knowledge about smut fungi has been derived from a few model smut pathogens, of which the best studied is *Ustilago maydis* (DC.), which causes gall formation on all aerial parts of its host plant *Zea mays* (L.) and its wild relatives Teosinte [3,4]. It shares with all smut fungi a parasitic dikaryotic phase and a saprophytic haploid phase during its life cycle. In its dikaryotic phase, the fungus grows inter- and intracellularly through the host tissue, suppresses the immunity, redirects the metabolic flow of the plant, and changes the development of its host. This is achieved by the secretion of effectors by the pathogen, which are likely hundreds of different molecules, including many secreted proteins [5]. Smut effector proteins are involved in host immune suppression, plant hormone signaling manipulation, developmental pathway redirection, and plant architecture modification [6,7,8,9,10,11,12,13,14,15,16]. In addition to *U. maydis*, only a few other characterized smuts are known to cause galls. *U. esculenta* forms smut galls restricted to the stem region of its host plant *Zizania latifolia* (Griseb.) [17], and *Melanopsichium pennsylvanicum* (Hirschhorn)*,* which infects dicot plants in the Polygonaceae family, induces galls in the inflorescence, leaves, and sometimes in the stems [18,19]. All gall-inducing smuts share the ability to influence cell division and cell expansion in their host plants in some way. The phytohormone auxin is a master regulator in plants for cell division and cell expansion and, therefore, is a key candidate for gall formation. Importantly, auxin biosynthesis genes are also found in smut genomes that do not cause galls during biotrophy, which strongly indicates a function of auxin beyond gall induction (Table 1). Some smut fungi species show greater similarity to *Ustilago maydis* auxin biosynthesis genes even though they are distantly related when comparing whole genomes (Figure 1)

This review discusses the current understanding of how auxin signaling in plants is used and manipulated by smut fungi and what one can learn from these fascinating biotrophic specialists. Due to space restrictions, some valuable work of colleagues was not given space in this review, for which the authors apologize. 

## 2. Auxin, the Master Regulator of Growth and Development in Plants

The word auxin originates from the Greek word “auxein”, which means “to grow”. Since its discovery in 1928 by Fritz Went [20], the roles of auxin have been implicated in almost all aspects of plant growth and development [21,22]. Auxin is involved in the control of processes such as cell division, elongation, differentiation, tropisms, flowering, apical dominance, lateral root formation, senescence, abscission, and responses to environmental stresses [23]. Indole-3-acetic acid (IAA) is the most prominent auxin and is widely investigated in plants, but other naturally occurring or synthetic molecules with auxin activity exist [24]. Auxin is synthesized primarily in the shoot meristem and young leaves from the amino acid tryptophan, but there are also tryptophan-independent pathways [25]. Once synthesized, auxin is transported in the plant through the phloem and a polar (directional) transport system [24]. Auxin concentrations in each tissue are tightly regulated by metabolism and polar transport [26], which, in turn, regulate auxin-mediated gene expression.

Auxin functions primarily by regulating gene expression through a pathway called canonical auxin signaling [26,27]. However, an increasing amount of data shows that auxin also functions through alternative mechanisms [28]. In this context, only a brief summary of canonical auxin signaling is presented, which has been highly conserved in plant evolution at least since the emergence of land plants [28]; noncanonical auxin signaling is reviewed elsewhere [22,28]. The core components of canonical auxin signaling include the auxin receptor of the TRANSPORT INHIBITOR RESPONSE1/AUXIN SIGNALING F-BOX (TIR1/AFB) family, the transcriptional repressors of the AUXIN/INDOLE-3-ACETIC ACID INDUCIBLE (Aux/IAA) family, and the transcription factors (TF) of the AUXIN RESPONSE FACTORS (ARF) family [21,26]. A TIR1/AFB F-box protein and an Aux/IAA transcriptional coregulator complex bind auxin, and this binding further promotes the interaction between TIR1/AFB and Aux/IAA, thus triggering ubiquitin-mediated degradation of Aux/IAA proteins via the proteasome (Figure 2). Aux/IAA proteins generally act as repressors to prevent auxin-responsive transcription by binding to ARFs [26,27]. The Aux/IAA proteins contain ethylene-responsive element binding factor-associated amphiphilic repression (EAR) motifs through which they recruit TOPLESS (TPL)/TOPLESS-RELATED (TPR) corepressors that can themselves recruit histone deacetylases (HDACs) to target loci; thus, they are responsible for the repression [21,29]. The cocrystal structure between TPL and the Aux/IAA EAR domain demonstrated that Aux/IAAs bind through the EAR motif to the so-called CTLH region at the N-terminus of the TPL repressor [30]. The degradation of Aux/IAA proteins leads to the derepression of ARFs, and auxin-responsive genes are thus activated. Auxin response factors (ARFs) are transcriptional factors that are bound to the cis-regulatory regions of auxin-responsive genes. ARFs can act as transcriptional activators or repressors depending on their function [31]. Numerous auxin-responsive genes, comprising early auxin-responsive genes (like the SAUR, IAA, and GH3 genes) and late auxin-responsive genes (such as genes involved in cell wall remodeling and growth), are expressed as a consequence of ARF activation [32]. Among these genes, a distinct set of genes is regulated depending on the cell type, indicating responses to auxin that are space- and time-specific. The complexity of these interactions increases with the duration of exposure to auxin [21].

## 3. Auxin in Plant–Pathogen Interactions

Plant defense responses are tightly regulated to optimize the tradeoffs between growth and defense [33]. Crosstalk between phytohormones plays a significant role in this regulatory process. In addition to the growth–defense antagonism, plants have further evolved a signaling antagonism in their response to pathogens of different lifestyles. Generally speaking, plants depend on Jasmonate (JA)/ethylene-dependent defense responses when defending against necrotrophic invaders, whereas salicylic acid (SA)-dependent signaling is key in fighting off biotrophic or hemibiotrophic pathogens during the phase when they depend on living host cells during colonization. Due to their different roles against various lifestyle dependencies of invading pathogens, SA and JA/Ethylene act antagonistic to each other. Nevertheless, JA/ethylene and SA hormones can be considered as clearly defense-related phytohormones. Auxin and other growth hormones in plants are modulators of defense responses, and auxin has been reported to mainly act antagonistically to SA signaling and vice versa [34,35]. Therefore, it is not surprising that the vast majority of reports of auxin signaling modulation promoting plant susceptibility come from biotrophic and hemibiotrophic plant–pathogen interactions [36,37,38]. Although not the focus of this review, the interaction-promoting role of auxins in symbiotic interactions of plants with mycorrhizal fungi, nodulating bacteria, or plant-growth-promoting rhizobacteria might also partly rely on the defense-modulating effects of auxin [39]. The strategies that microbes have evolved to manipulate auxin signaling are fascinatingly versatile. Various plant-associated microbes produce and secrete auxins themselves [40], while others manipulate their distribution. Auxin is redistributed by PIN-FORMED, ABC, and AUX/LAX transporters within the plant body to shape morphogenesis. The relocalization of PIN transporters has been observed during cyst nematode infection, which likely causes an influx of auxins into the initial syncytial cell and is also involved in the expansion of the nematode-feeding structure [41]. The overexpression of the potyvirus suppressor of RNA silencing, HC-Pro, in the model plant *A. thaliana* (L.) led to the downregulation of DNA methylation in the regulatory region of the YUCCA genes of the indole-3-pyruvic acid (IPyA) auxin biosynthesis pathway. As a consequence, changes in the methylation pattern lead to transcriptional upregulation and, therefore, increase auxin biosynthesis *in planta* [42]. *Pseudomonas syringae* (Van Hall) have been shown to secrete AvrRpt2 to interfere with auxin-signaling transcriptional repressor Aux/IAA proteins to increase auxin signaling for disease progression, as also shown for viral effectors [38,43,44,45]. The corepressor TOPLESS is manifold-manipulated by *U. maydis,* which will be described in more detail below. In contrast, the auxin response factor TF is targeted by various rice-infecting viruses. These viruses encode effectors, which block the dimerization of OSARF17 or, in other cases, direct DNA binding and thus render this specific auxin response factor inactive to block the transcription of antiviral gene responses under its control [46]. Similarly, several plant pathogenic bacteria encode IAA-Lys synthases. IAA-Lys is a less active form of IAA. The bacterial pathogen of oleander, *Pseudomonas savastanoi pv. neri*, produces IAA-LYS synthases and seems to control the free levels of IAA in infected tissue in this way [47]. There are not many reports in which auxin has been shown to play a general promoting role in plant immunity. In the fungal *Rhizoctonia solani* (J.G. Kühn)/rice interaction, external auxin treatment increased resistance [48]. Mutants of *Arabidopsis thaliana* (L.) affected in the auxin pathway are shown to be more susceptible than wild-type plants to the necrotrophic fungus *Alternaria brassicicola* (Schwein.). Upon infection, plants respond with the upregulation of auxin responses. The cotreatment of plants with both MeJA and IAA leads to the synergistic upregulation of typical JA marker genes [49]. This finding supports the hypothesis that the positive role of auxin signaling against necrotrophs might be through the enhancement of JA signaling.

## 4. Auxin Biosynthesis in Smut Fungi

Many plant pathogens possess the capacity to synthesize indole-3-acetic acid (IAA), the major form of auxin in plants. However, so far, the knowledge of auxin biosynthesis in smut fungi has been largely derived from the model smut fungus *U. maydis*. IAA is known to be synthesized through several pathways; the most common route is the conversion of L-tryptophan to IAA, but there are also tryptophan-independent pathways [50]. Although both pathways coexist in some organisms, most studies have focused on tryptophan-dependent pathways. At least six different tryptophan-dependent biosynthetic pathways have been described according to their key intermediates: indole-3- acetamide (IAM), indole-3-acetaldoxime (IAOx), indole-3-pyruvic acid (IPyA), indole-3-acetamide (IAM), tryptamine (TAM) and indole-3-acetonitrile (IAN), and Trp side chain oxidase (TSO) [51,52].

The gall-forming corn smut fungus *U. maydis* has been reported to produce indole acetic acid, which is also associated with elevated auxin levels during its biotrophy in maize [53,54]. Several genes involved in the IAA biosynthesis pathway from *U. maydis* have been identified and characterized [55,56], including two NAD-dependent IAAld dehydrogenases (*iad1* and *iad2*) and two predicted aromatic amino acid aminotransferase genes (*tam1* and *tam2*). Aminotransferases Tam1 and Tam2 participate in the conversion of tryptophan to IPyA. Downstream of this, the dehydrogenases Iad1 and Iad2 are shown to be involved in the formation of IAA from indole-3-acetamide (IAAld). The increase in host IAA levels upon *U. maydis* infection was significantly reduced in tissue infected with quadruple Δiad1Δiad2Δtam1Δtam2 *U. maydis* mutants, suggesting a critical contribution of fungal IAA production to elevated IAA levels in infected tissue. However, a reduction in fungal auxin production did not have any effect on gall formation and the development of host plant disease symptoms, suggesting alternative strategies for this phenomenon [56]. In the genome of the head smut fungus *Sporisorium scitamineum* (Syd.), a gene coding for the tryptophan aminotransferase *SsAro8* has been identified. The tryptophan aminotransferase SsAro8 has been shown to catalyze the first step of tryptophan-dependent IAA production, but also for the Ehrlich pathway for tryptophol. It was found to be essential for mating/filamentation and proper biofilm formation in *S. scitamineum* [57]. As SsAro8 is at the beginning of several pathways with different intermediates and products, it is not clear whether any of the fungal phenotypes identified are related to altered IAA production or due to the lack of other Aro8-dependent metabolites. Furthermore, in *U. scitaminea* and *U. esculenta*, the ability to synthesize IAA from tryptophan was found through the indole-pyruvate (IPA) pathway. *U. esculenta* apparently could convert IAAld and indole-lactic acid (ILA) into IAA in addition to tryptophan [58]. The finding that auxin biosynthesis genes are expressed in a carbon-source-dependent manner in axenic culture in *U. maydis* [56], but also findings in other fungi, like the ascomycete *M. oryzae* (T.T. Hebert), show that IAA acts as a quorum-based modulator of virulence [59]. These findings raise questions about whether, beyond its obvious impact as a phytohormone, fungal auxin possibly also plays a role as an intraspecific signaling molecule of IAA-producing smuts. 

## 5. Transcriptional Reprogramming of Auxin Signaling by Smuts

Auxin, as a central plant growth hormone, has long been suspected to be key in the gall formation induced by *U. maydis*. That is why it came as a surprise when Reineke et al. (2008) [56] provided genetic evidence that fungal IAA biosynthesis mutants still induce normal gall formation. On the other hand, transcriptomic time course experiments on maize seedling infection with *U. maydis* showed transcriptional upregulation of auxin-regulated genes, including several auxin response factor genes and auxin efflux carrier genes of maize [60]. Systematic functional effector screens revealed years later that the central corepressors of the auxin signaling pathway, TOPLESS (TPL) and TOPLESS-related (TPRs) proteins, were negatively modulated in their function by *U. maydis* effector proteins. This functional redundancy with fungal-derived auxin secretion might well be the reason why the IAA biosynthesis mutants of *U. maydis* are not impaired in virulence. The study on the *U. maydis* effector Naked1 (Nkd1) revealed that TPL is a positive regulator of PAMP-triggered immunity [8]. The auxin-signaling-inducing activity of Nkd1 expressed *in planta* was directly correlated with its ROS burst-suppressing activity triggered by PAMP, and both were linked to its ability to bind TPL. The ERF-associated amphiphilic repression (EAR) motif at its C-terminus was identified to be crucial for its ability to bind to TPL. It could be functionally replaced with respect to PTI-suppression and DR5 (auxin signaling reporter) induction in transient expression assays of the respective Nkd1 fusion constructs in *Nicotiana benthamiana* (Domin) by the strongly modified EAR repressor motif SRDX [61].

Importantly, although the Nkd1-SRDX fusion protein shows stronger binding to TPL and also equally good induction of auxin signaling and PTI-suppressive activity when used as a complementation construct under its endogenous promoter in Δ*nkd1 U. maydis* deletion strains. However, it did not complement the virulence defect and even severed it. Furthermore, transcriptomes of *A. thaliana* plants expressing either Nkd1 or Nkd1-SRDX indicate that the genes differentially regulated by the two protein versions differ significantly [8]. This finding implies that the native Nkd1 protein has an EAR motif that interferes only to some extent with the functions of TPL and, thus, does not inhibit all of its functions. This interference is possibly a strategy to avoid pleiotropic, adverse effects on the host’s viability, an evolutionary driving force for any biotroph, including the smut *U. maydis*. This specificity in interference with the host protein TPL is also underpinned by the finding of the Jasmonate/ethylene signaling inducer 1 (Jsi1) effector. This effector of *U. maydis* has also been shown to interact with TPL, but, in this case, with a C-terminal WD40 domain and an alternative EAR motif, thus triggering major changes in ERF pathway-related transcripts upon *in planta* expression, a pathway coregulated by Jasmonate and ethylene signaling [14].

Specificity in binding was also discovered for a whole set of TPL-interacting protein effectors, the so-called Tip effectors of *U. maydis*, of which, in the meantime, eight effectors have been described. Tip1-5 are found in the genomic cluster 6a, which encodes a total of eight secreted proteins and is syntenic to clusters 6–10 in the smut fungus *S. reilianum*. Interestingly, Tip1-5 do not have any recognizable EAR motifs in their sequence but have been shown to bind to the EAR-binding pocket of TPL proteins in Y2H assays. As in the case of Nkd1, the tip cluster effectors also showed various binding capacities to the different TPL/TPR proteins of maize, indicating specificity [11]. A systematic Y2H screening among nearly 300 candidate effectors with the RELK2 maize TPR protein led to the discovery of Tip6-8 [13]. Tip6 and Tip7 have two and one EAR motifs, respectively, in their sequence, whereas the Tip8 sequence does not contain any known EAR motifs [13]. Analogous to the Tip1-5 cluster, the newly identified Tip6-8 effectors also interact with the N-terminal part of TPL proteins. In an independent parallel study, Tip6 was found to interact with the N-terminus of RELK2 through its two EAR motifs, and mutation in the motif eliminates its ability to restore the virulence defect of Tip6 deletion mutants. Transcriptomic comparisons on maize infected either with the solopathogenic *U. maydis* strain SG200 or a Δtip6 mutant strain revealed that a set of transcripts encoding for transcription factors of the AP2/ERF B1 subfamily changed significantly, indicating that the presence of Tip6 is necessary for their transcriptional modulation by *U. maydis* [15].

Although the literature suggests that most auxin signaling manipulation by *U. maydis* is achieved via TPL interference, there is likely more to be discovered. First of all, even the octuple deletion mutant of TPL-interacting effectors shows only a moderate reduction in virulence, a possible indication of functional redundancy that is not covered [13]. Furthermore, cluster 6a also contains, besides Tip1-5, the effector Umag_11416, which is also auxin-signaling-inducing but does not interact with TPL and could lead to the discovery of alternative ways of growth hormone signaling manipulation in smuts in the future [11] (Figure 2). One important role of auxin in plants is the suppression of branching, called apical dominance, caused by a gradient of auxin that is highest in the apical meristem. For this function, auxin transporters play a crucial role. During colonization, the head smut fungus *Sporisorium reilianum* (J.G. Kühn) causes many other smuts to change the inflorescence and branching architecture of its host plant. One effector that, upon deletion, leads to a loss of branching is the effector Suppressor of Apical Dominance 1 (SAD1). SAD1-GFP accumulates upon expression *in planta*, mainly within the nucleus of the plant. Real-time PCR on maize infected either with wild-type *S. reillianum* or Δsad1 mutant strain shows an increase in the transcript levels of the auxin transporter PIN-FORMED1 in the root and a reduction in the branching regulator TEOSINTE BRANCHED1 in the stalk. How exactly SAD1 acts to induce these important changes awaits further research [6].

## 6. Plants’ Counter-Defense to Auxin Signaling Manipulation?

Both programmed cell death and auxin (as a key growth hormone) are important for the development of plants. On the one hand, auxin is described as negatively acting on cell death processes. On the other hand, there are examples in which auxin promotes plant cell death, possibly by promoting ethylene biosynthesis [62]. Interestingly, in the case of several TPL-interacting effectors *in planta*, overexpression leads to cell death responses. Among them is the ethylene-signaling-inducing Jsi1 effector, but also the auxin-signaling-inducing Nkd1 effector when fused to the very strong TPL-binding SRDX EAR motif. An interesting observation was made here that, although SRDX fusion to Nkd1 strongly inhibited PAMP-triggered ROS burst responses, transcriptomic analysis of these plants indicated a strong upregulation of SA-related PR genes, R genes including the disease resistance gene SNC1, the TIR-NBS-LRR resistance protein RLM1, and also the plasma membrane-localized NADPH-oxidase RBOHD, which is needed for the PAMP-triggered ROS burst [8]. This finding implies that the PTI-promoting role of TPL is likely post-transcriptionally regulated because even elevated transcriptional levels of the NADPH-oxidase do not overrule the activity of the *U. maydis* effector in suppressing PAMP-triggered ROS burst responses via interaction with TPL. For *U. maydis*, but considering the orthologs of several TPL-interacting effectors in other smuts, TPL is an effector hub. Furthermore, in other pathosystems, TPL has also been shown to interact with effectors. The *Fusarium oxysporum* (Schlecht.) effector Six8 has been identified to interact with TPL proteins that cause cell death in an SNC1-dependent manner [63]. The effector PopP2 of *Ralstonia solanacearum* (Smith 1896) has an EAR motif, which is associated with PTI-suppressive activity and avirulence effects in the plant immune system [64]. In the case of Nkd1-SRDX overexpressing plants, the transcriptome and the cell death that appears about three days after the induction of Nkd1-SRDX support the idea that TPL is guarded by the plant defense system and that its over-manipulation ultimately leads to cell death, even though auxin signaling is induced. This assumption can be well-explained with the coevolutionary principle of the ZigZag model. It it likely that this protection occurs via a guard process on TPL-activity and not by direct effector binding because respective *U. maydis* effectors overexpressed either in *A. thaliana* or *N. benthamiana* (both of which are not host plants of this smut) showed effector-triggered cell death induction. As *U. maydis* does not infect these dicot plants, it would be surprising if these nonhost plants would bear specific resistance genes for the recognition of, for example, Jsi1, Nkd1, and Tip8, to mention just a few. Why a biotrophic smut carries in its effectome so many effectors with the potential to cause cell death is unclear but this opens up space for speculation. One obvious reason is that, upon heterologous overexpression, some of the TPL-interacting *U. maydis* effectors reach much higher cellular concentrations than those under natural infection conditions, and this leads to unwanted “over-manipulation”. Additionally, it can be expected that smut effectomes carry effectors involved in PCD suppression, thereby compensating for potential effector activity recognition. 

## 7. Conclusions and Outlook

Biotrophic pathogens, including smut fungi, evolved strategies to employ plant signaling pathways in their favor. Growth signaling pathways, including auxin, are especially relevant, as hardwired antagonisms between growth and defense signaling pathways in plants can, in this way, be exploited. While the direct targeting of immune components by effectors might lead to fast co-evolving receptors on the host defense side, the subtle activation of essential growth pathways might be far more difficult to detect without causing false alarms during growth and development. Possibly due to this crucial importance, *U. maydis*, and likely many other smuts, evolved numerous factors and strategies to activate auxin signaling. Whether this occurs in a cell type or tissue-specific manner or is rather constitutive is so far unclear. The identified mechanisms in host auxin signaling manipulation in smuts are on the level of fungal-derived auxin and transcriptional control of auxin-responsive genes. However, it would not be surprising if, at the level of the transport or stability of auxin, points of interference might be exploited; future research might provide answers for this question.

## Figures and Tables

**Figure 1 jof-09-01184-f001:**
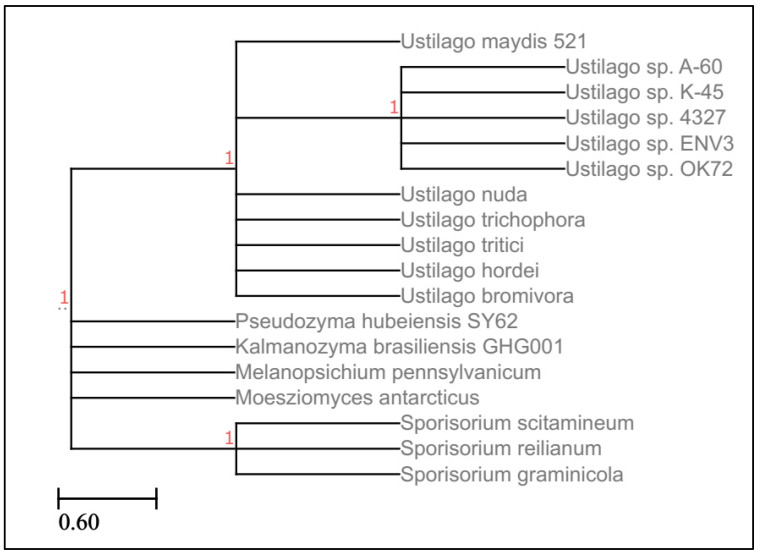
Phylogenetic tree based on whole genome comparison among smut fungi created using Lifemap from NCBI and viewed using ETEToolkit. Support values are shown in red.

**Figure 2 jof-09-01184-f002:**
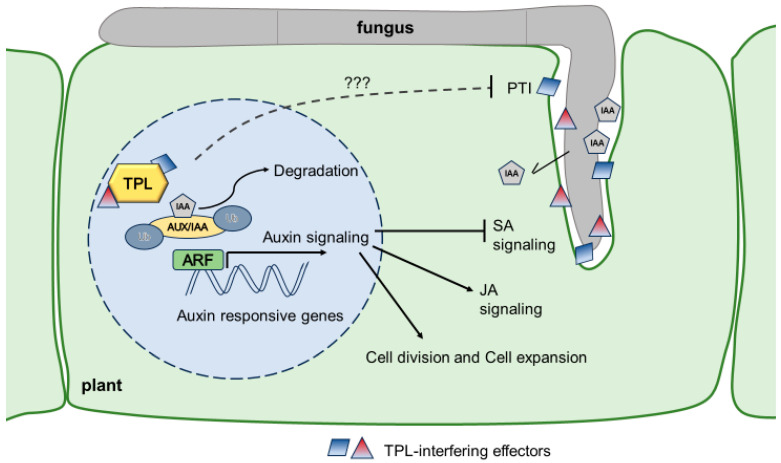
Model how *U. maydis* influences auxin signaling in maize. *U. maydis* produces IAA and releases it during the colonization of the host tissue. Additionally, it secretes at least ten translocated effectors (Tip1-8, Nkd1, Jsi1) which all interact with members of the TPL/TPR protein family and, among others, also lead to the derepression of auxin signaling and PTI suppression through unknown mechanisms (???) Figure 2 was created using Microsoft PowerPoint.

**Table 1 jof-09-01184-t001:** Sequence comparison of auxin biosynthesis genes from different smut fungi. The table shows the percentage similarity at the protein level of *Tam1*, *Tam2*, *Iad1*, and *Iad2* across different smut fungi compared to *U. maydis*. All four genes are conserved among smut fungi regardless of gall-forming species, indicating that these auxin biosynthesis genes possibly have a supporting function in gall induction, assuming functional conservation. The colors are based on a 3-color scale in Excel. The cells that hold the minimum values are colored red, the median values are colored yellow, and the cells with maximum values (100) are colored green. All other cells are colored proportionally.

		Percentage Identity			
Scientific Name	Gall Formation	*Tam1*	*Tam2*	*Iad1*	*Iad2*
*Ustilago maydis* 521	+	100	100	100	100
*Pseudozyma hubeiensis*	-	85.53	91.05	95.98	90.66
*Sporisorium scitamineum*	-	86.17	89.86	96.38	37.15
*Sporisorium graminicola*	-	86.32	89.26	96.78	87.55
*Sporisorium reilianum*	-	83.58	89.46	95.17	36.52
*Ustilago* sp., UG-2017a	-	80.13	84.26	94.57	65.49
*Ustilago hordei*	-	79.92	84.46	93.96	36.94
*Ustilago trichophora*	+	82.24	89.46	96.18	84.41
*Ustilago nuda*	-	79.96	84.06	93.76	36.73
*Ustilago tritici*	-	79.50	83.47	93.96	59.63
*Moesziomyces antarcticus*	-	77.12	84.69	94.16	85.86
*Melanopsichium pennsylvanicum*	+	74.23	87.08	94.57	36.17
*Kalmanozyma brasiliensis* GHG001	-	74.04	86.48	92.96	82.37
*Ustilago bromivora*	-	74.27	83.86	94.57	67.98

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
