# Peer review of "Manipulation of Auxin Signaling by Smut Fungi during Plant Colonization"

_jof, 2023, doi:10.3390/jof9121184_

Round 1

Reviewer 1 Report

Comments and Suggestions for Authors

The authors emphasize smut fungi in the title of this manuscript. However, the contents of ‘Auxin - The Master Regulator of Growth and Development in Plants’, ‘Auxin in plant-pathogen interactions’, and ‘Plants Counter-defense to Auxin signaling Manipulation’ are general in the interaction between plants and fungal pathogens. Therefore, the topic shown in the title does not match up with the contents of this manuscript. Furthermore, most of the contents related to smut fungi are results of U. maydis. What about other smut fungi.? Thus, the authors should find a new perspective on this manuscript, since there are many reviews about IAA in plants and fungi, and studies on IAA in smut fungi are almost focused on U. maydis.

 1Figure 1A is a table! It should be individually named as Table 1.

2. The contents of Figure 1 are not described in the whole manuscript. Why did the authors choose Tam1, Tam2, Iad1, and Iad2 for similarity analysis? What does the similarity mean? What information did the authors want to show us through the phylogenetic tree in Figure 1B? Did the phylogenetic tree build with the whole genome of each strain?

Comments on the Quality of English Language

The English of the manuscript should be improved before resubmission.

1. Line 9: Change ‘in order to’ to ‘to’

2. Line 11: ‘as one of the largest groups in basidiomycetes’ is better.

3. Line 16: ‘Smut fungi, their development, and disease symptoms’, a comma before ‘and’ is missing. Please check the full manuscript.

4. Line 39: Change to ‘only a few’.

5. Line 43: Change ‘the other’ to ‘another’.

6. Line 116: ‘roles’

7. Line 119: delete ‘rather’

8. Line 125: delete ‘too’

9. Line 128: ‘manipulate’

10. Line 154: ‘leads’

11. Line 172: ‘include’

12. Line 199: ‘signaling’

13. Line 222: ‘strongly’

14. Line 250: delete ‘also’

15. Line 281: ‘in the case’

16. Lines 284-285: ‘an interesting’

17. Line 290: ‘implies’

18. Line 322: Change ‘be detected’ to ‘detect’

Reviewer 2 Report

Comments and Suggestions for Authors

The review carried out is very interesting and up-to-date, it is only necessary to remember that the articles are written in the third person, not in the first person and that the scientific names of both plants and pathogens carry descriptors.

Comments on the Quality of English Language

Reviewer 3 Report

Comments and Suggestions for Authors

Review remarks on the review manuscript entitled “Manipulation of auxin signaling by smut fungi during plant colonization” submitted by Nagarajan et al. to JoF-MDPI journal. The authors summarized the role of auxin signaling variations by fungal smut during the colonization process.  Auxin is an important hormone for proper plant growth and development, and its signaling strongly associated with plant development and immunity responses. Smuts as one of the largest groups of basidiomycetes.

The present article is well-written and discussed systematically.

 Overall comments:

The authors have discussed an important and interesting issue. Scientifically, the MS is strong, and I recommend its publication in the “JoF-MDPI journal” after minor revision. English is well presented, but some sentences need improvement entire the MS. Even some sentences are difficult to understand.

Some specific comments are as stated below:

Specific Comments:

Abstract: Please incorporate some more relevant/ key points/message of the proposed article

Keywords are missing, plz incorporate it

The introduction section is missing or started from line 16??

Conclusion section should be improved

I think the references have some issues; please check them carefully as per journal guidelines

Congratulations to all authors on writing such a fine piece of work!

Comments on the Quality of English Language

English is well presented, but some sentences need improvement entire the MS

Round 2

Reviewer 1 Report

Comments and Suggestions for Authors

 Accept in present form

Comments on the Quality of English Language

 Accept in present form